



**Derivation of aerosol fluorescence and water vapor Raman depolarization ratios from lidar**
**measurements**
Igor Veselovskii[1], Qiaoyun Hu[2], Philippe Goloub[2], Thierry Podvin[2], William Boissiere[2], Mikhail
Korenskiy[1], Nikita Kasianik[1], Sergey Khaykyn[3], Robin Miri[2]
*[1]Prokhorov General Physics Institute of the Russian Academy of Sciences, Moscow, Russia.*
*[2]Univ. Lille, CNRS, UMR 8518 - LOA - Laboratoire d'Optique Atmosphérique, F-59650 Lille,*
*France*
*[3]Laboratoire Atmosphère Observations Spatiales, UVSQ, CNRS, Sorbonne University,*
*Guyancourt, France*
**Correspondence**: Qiaoyun Hu (qiaoyun.hu@univ-lille.fr)
**Abstract**
Polarization properties of the fluorescence induced by polarized laser radiation are widely
considered in laboratory studies. In lidar observations, however, only the total scattered power of
fluorescence is analyzed. In this paper we present results obtained with a modified Mie-Raman-
Fluorescence lidar operated at the ATOLL observatory, Laboratoire d'Optique Atmosphérique,
University of Lille, France, allowing to measure depolarization ratios of fluorescence at 466 nm
($\delta_F$) and of water vapor Raman backscatter. Measurements were performed in May-June 2023
during Alberta forest fires season when smoke plumes were almost continuously transported over
the Atlantic Ocean towards Europe. During the same period, smoke plumes from the same sources
were also detected and analyzed in Moscow, at General Physics Institute (GPI), with a 5-channel
fluorescence lidar able to measure fluorescence backscattering at 438, 472, 513, 560 and 614 nm.
Results demonstrate that, inside the boundary layer (BL), urban aerosol fluorescence is maximal
at 438 nm, then it gradually decreases with wavelength. Results also show that the maximum of
the smoke fluorescence spectrum shifted towards longer wavelengths. The smoke layers observed
within 4-6 km present a maximum of fluorescence at 513 nm while, in the upper troposphere (UT),
the maximum shifts to 560 nm. Regarding fluorescence depolarization, its value typically varies
inside the 45-55 % range, however several smoke plume layers detected above 10 km were
characterized by a $\delta_F$ increasing up to 70%. Inside the BL, the fluorescence depolarization ratio
was higher than that of smoke and varied inside the 50-70% range. Moreover, in the BL, $\delta_F$ appears





to vary with atmospheric relative humidity (RH) and, in contrast to the elastic scattering,
fluorescence depolarization increases with RH.

The depolarization ratio of the water vapor Raman backscattering is shown to be quite low

(2±0.5%) in the absence of fluorescence, because the narrowband interference filter in the water
vapor channel selects only strongest vibrational lines of the Raman spectrum. As a result,
depolarization of the water vapor Raman backscattering is sensitive to the presence of strongly
depolarized fluorescence backscattering. The fluorescence contamination into the water vapor
Raman channel can be calculated from the water vapor Raman depolarization ratio with the only
assumption that $\delta_F$ remains constant within the 408-466 nm range.

**1. Introduction**

Possibility to measure the laser induced fluorescence becomes an important added-value to

existing Mie-Raman lidars, because fluorescence measurements provide new independent
information about aerosol properties. Nowadays, the spectroscopic lidars based on 32-channel
PMT combined with spectrograph proved the ability to measure the fluorescence spectrum
(Sugimoto et al., 2012; Reichardt, 2014; Reichardt et al., 2018, 2023, Liu et al., 2022). On the
other hand, the lidars, with a single fluorescence channel can be widespread due to their simplicity
(Rao et al., 2018; Veselovskii et al., 2020). Such single channel fluorescence measurements
combined with depolarization measurements at elastic wavelength, provide new independent
information about aerosol type (Veselovskii et al., 2022; Wang et al., 2023). However, in all lidar
studies, only total scattered power was analyzed, while polarization properties of the fluorescence
were ignored. At the same time, fluorescence depolarization measurements are widely used in
laboratory research (Lakowicz, 2006). When polarized laser radiation is used for excitation, the
fluorescence emission is also partly polarized and degree of emission polarization (anisotropy)
depends on the fluorescence lifetime, on the angle between excitation and emission dipoles, and
on the rotational mobility of molecules (Lakowicz, 2006). In fluorescence spectroscopy the
anisotropy is introduced as given by Eq.1:
$$r = \frac{P_F^{\parallel} - P_F^{\perp}}{P_F^{\parallel} + 2P_F^{\perp}} \qquad\qquad (1)$$
Here $P_F^{\parallel}$ and $P_F^{\perp}$ are the powers of co- and cross-polarized fluorescence components. In lidar
measurements, however, a commonly used parameter is the depolarization ratio, $\delta_F$.



$$\delta_F = \frac{P_F^{\perp}}{P_F^{\parallel}}$$ (2)
Anisotropy and depolarization ratio are two interconvertible parameters
$$r = \frac{1 - \delta_F}{1 + 2\delta_F}$$ (3)
For randomly oriented fluorophores with collinear absorption and emission dipoles, in the
absence of rotational motion, the anisotropy is r=0.4 (Lakowicz, 2006), which corresponds to
$\delta_F$=33%. This is the minimal value one can expect in lidar measurements. Existence of the angle
between absorption and emission dipoles, as well as molecule rotation in the process of emission
will increase $\delta_F$. Thus, measurement of fluorescence depolarization ratio may bring additional
information about atmospheric aerosol. Moreover, depolarization measurements help to analyze
the impact of fluorescence on water vapor Raman measurements.
Water vapor is a key atmospheric component playing essential role in the planet's radiative
balance, and Raman lidars today are widely used for vapor observations (Whiteman, 2003, Chouza
et al., 2022 and references therein). However, when the UV laser beam passes through a smoke
layer, the broadband fluorescence signal is induced and its spectrum includes the region of water
vapor Raman lines. Thus, the signal in the water vapor channel (around 407.5 nm, when 354.7 nm
radiation is used for stimulation) becomes contaminated by the fluorescence backscatter signal
(Immler et al.. 2005; Immler and Schrems, 2005). This contamination can be reduced by
decreasing the width of the transmission band in the water vapor channel down to tenths of nm.
However, as it was shown recently, fluorescence still remains the issue, especially inside the smoke
layers in high troposphere (Chouza et al., 2022; Reichardt et al., 2023).
Depolarization measurements provide an opportunity to monitor the presence of
fluorescence signal in Raman channel. The Q-branch of water vapor Raman lines (near 407.5 nm)
provides a weakly depolarized backscatter, while fluorescence is strongly depolarized. Thus the
presence of fluorescence should increase the depolarization ratio of signal in the water vapor
channel. Moreover, if depolarization ratios of water vapor and fluorescence are known, the
contribution of fluorescence to the measured water vapor mixing ratio (WVMR) can be evaluated.
In this article, report and analyze, for the first time, depolarization ratio of aerosol
fluorescence and depolarization ratio of water vapor Raman backscatter from lidar observations
performed at the ATOLL observatory (ATmospheric Observation at liLLe), Laboratoire





d'Optique Atmosphérique, University of Lille, during dense smoke events that occurred in May -
June 2023. We start with a description of the experimental setup in Sect.2.1 and derive, in Sect.
2.2, the main equations for estimating the fluorescence contribution to the water vapor Raman
channel. In the first part of the results section (Sect.3.1), the fluorescence depolarization ratio over
ATOLL is analyzed for different aerosol types. The measurements of fluorescence spectra
performed with a new five-channel fluorescence lidar, operated in Moscow, are presented in
Sect.3.2. In Sect. 3.3, we analyze depolarization in the water vapor Raman channel and estimate
the contamination of fluorescence to the derived WVMR. The article ended with a conclusion.

**2.   Experimental setup and data analysis**
*2.1 Lidar system*
In our study, two lidar systems were considered. The first one, LILAS ((LIlle Lidar
AtmosphereS) is a multiwavelength Mie-Raman-Fluorescence lidar, whereas the second one is a
multiwavelength fluorescence lidar operated by General Physics Institute (GPI), Moscow (Veselovskii
et al., 2023). Both systems are based on a tripled Nd:YAG laser (Q-Smart 450) with a 20 Hz
repetition rate and pulse energy about 100 mJ at 355 nm. Backscattered light in both systems is
collected by a 40 cm aperture telescope and the lidar signals are digitized with Licel transient
recorders with 7.5 m range resolution, allowing simultaneous detection in the analog and photon
counting mode.
LILAS   allows   the   so   called   $3\beta+2\alpha+3\delta$   configuration,   including   three   particle
backscattering ($\beta_{355}$, $\beta_{532}$, $\beta_{1064}$), two extinction ($\alpha_{355}$, $\alpha_{532}$) coefficients along with three particle
depolarization ratios. The Raman channel with 407.54/0.3 nm interference filter allows also water
vapor profiling. At the end 2019, the lidar was modified to enable fluorescence measurements. A
part of the fluorescence spectrum is selected by a wideband interference filter of 44 nm width
centered at 466 nm (Veselovskii et al. 2020).
In the fluorescence lidar of GPI only 355 nm wavelength is emitted, while fluorescence is
measured in five spectral intervals. The central wavelengths and widths of transmission bands are:
438(29), 472(32), 513(29), 560(40) and 614(54) nm (Veselovskii et al., 2023). Thus, the
fluorescence spectrum could be sampled. At GPI, the measurements were performed at an angle
of 48 deg to the horizon. The strong sunlight background restricts the fluorescence observations
of both systems to nighttime hours.



Several properties can be derived from fluorescence. The fluorescence backscattering
coefficient, $\beta_{F\lambda}$, at wavelength $\lambda_F$, is calculated from the ratio of fluorescence and nitrogen Raman
backscattering signals, as described in Veselovskii et al. (2020). We remind that $\beta_{F\lambda}$ is related to
fluorescence signals integrated over the filter transmission band $D_\lambda$. In Moscow measurements are
performed at five wavelengths, and to compare $\beta_{F\lambda}$ between different channels one makes use of
the "fluorescence spectral backscattering coefficient" $B_\lambda = \dfrac{\beta_{F\lambda}}{D_\lambda}$ (fluorescence backscattering per
spectral interval). LILAS has only one single fluorescence channel, therefore, when presenting
data from LILAS, for the sake of simplicity, one uses notation $\beta_{F466}=\beta_F$. The intensive property
characterizing aerosol fluorescence is the fluorescence capacity $G_{F\lambda}$, which is the ratio of the
fluorescence backscattering at wavelength $\lambda_F$ to backscattering coefficient at laser wavelength
$G_{F\lambda} = \dfrac{\beta_F}{\beta_\lambda}$. This ratio, in principle, can be calculated for any laser wavelength. For LILAS
observations $G_{F\lambda}$ is calculated with respect to $\beta_{532}$, because $\beta_{532}$ is derived with rotational Raman
scattering and is considered to be the most reliable. And again, when presenting LILAS data, for
simplicity one will use notation $G_{F\lambda}=G_F$. In this work, all profiles of aerosol properties are
smoothed with the Savitzky – Golay method, using second order polynomials with 8 points in the
window.
Additional information about the atmospheric thermodynamic state was available from
radiosonde measurements performed at Herstmonceux (UK) and Beauvechain (Belgium) stations,
located 160 km and 80 km away from the ATOLL observatory respectively. When calculating the
relative humidity, one then used the water vapor profiles measured by Raman lidar and temperature
profiles provided by the radiosonde.
As discussed in the introduction, measurements of fluorescence depolarization ratio and
depolarization of water vapor Raman backscatter are expected to bring new information about
aerosol properties and fluorescence contamination in the water vapor Raman channel. In 2023,
LILAS was upgraded to allow depolarization measurements at both 466 nm and 408 nm.
Corresponding optical layout is shown in Fig.1. Dichroic mirrors DM separate 387, 408 and 466
nm components, while polarizing cubes split the components with polarizations oriented parallel
(s) and perpendicular (p) to the laser polarization. For both channels the polarizing cube PBS251
from ThorLabs was used. Fluorescence depolarization ratio, $\delta_F$, and water vapor Raman scattering



depolarisation ratio, $\delta w$, are both defined and calculated as a ratio of the perpendicular to the
parallel respective components. Calibration was performed as described in Freudenthaler et al.
(2009). The uncertainty of calibration is estimated to be below 15% for both 466 and 408 nm
channels.

**2.2 *Expressions for estimating fluorescence impact on water vapor measurements*.**
As discussed in the recent work from Chouza et al. (2022) and Reichardt et al. (2023), the
broadband aerosol fluorescence is expected to contribute to the signal measured in water vapor
Raman channel. Below, one provides the basic equations for estimating this contribution, based
on the measurements of the depolarization ratio in the water vapor Raman channel. The
backscattered radiative power, at the laser wavelength $\lambda_L$, from distance $z$, can be modeled, after
background subtraction, by
$$P_L = O(z)\frac{1}{z^2}C_L \beta T_L^2 \qquad (4)$$
Here $O(z)$ is the geometrical overlap factor, which is assumed to be the same for all channels. $C_L$
is the range independent constant, including efficiency of the detection channel. $T_L$ is one-way
transmission, describing light losses on the way from the lidar to distance $z$ at wavelength $\lambda_L$.
$$T_L = \exp\left\{-\int_0^z [\alpha^a(\lambda_L, z') + \alpha^m(\lambda_L, z')]dz'\right\} \qquad (5)$$
Backscattering and extinction coefficients contain aerosol ($a$) and molecular ($m$) contributions:
$$\beta_{\lambda_L} = \beta_{\lambda_L}^a + \beta_{\lambda_L}^m \text{ and } \alpha_{\lambda_L} = \alpha_{\lambda_L}^a + \alpha_{\lambda_L}^m.$$
Radiative power in nitrogen Raman, water vapor Raman, and fluorescence channels can be
written in a similar way.
$$P_R = O(z)\frac{1}{z^2}C_R \sigma_R N_R T_L T_R \qquad (6)$$
$$P_W = O(z)\frac{1}{z^2}C_W N_W \sigma_W T_W T_L \qquad (7)$$
$$P_F = O(z)\frac{1}{z^2}C_F \beta_F T_F T_L \qquad (8)$$
Here $C_R, C_W, C_F$ are the corresponding range independent constants. Terms $T_R, T_V,$ and $T_F$ are one-
way transmissions at wavelengths $\lambda_R, \lambda_W, \lambda_F$, corresponding to the centers of transmission bands of





the channels. Terms $N_R$ and $N_W$ are the concentrations in nitrogen and water vapor molecules while
$\sigma_R$, $\sigma_W$ are their Raman differential scattering cross sections respectively. The fluorescence
backscattering coefficient, $\beta_F$, is introduced the same way, as described in (Veselovskii et al.,

2020).

The power of fluorescence signal that leaks to the water vapor channel is:
$$P_{FW} = O(z)\frac{1}{z^2}C_W\beta_{FW}T_W T_L \tag{9}$$
Here $\beta_{FW}$ is fluorescence backscattering at wavelength $\lambda_W$. The WVMR, $n_W$, can be obtained from
Eq.6 and Eq.7, if the calibration constant $K_W = \dfrac{C_R}{C_W}\dfrac{\sigma_R}{\sigma_W}$ is known and is given by Eq. 10
$$n_W = K_W\frac{P_W}{P_R}\frac{T_R}{T_W} \tag{10}$$
The fluorescence backscattering coefficient, $\beta_F$, derived from Eq.6 and Eq.8, also contains
the calibration constant $K_F$. The procedure of calibration is described in Veselovskii et al. (2020).
Finally, $\beta_F$ is given by Eq. 11.
$$\beta_F = K_F n_R\frac{P_F}{P_R}\frac{T_R}{T_F} \tag{11}$$
Here $n_R = \dfrac{N_R(z)}{N_R(z=0)}$ is relative change of number density of nitrogen molecules with height.
The fluorescence signal in the water vapor channel can be expressed from $P_F$ using parameter $\eta$,
which depends on ratio of fluorescence cross sections at wavelengths $\lambda_W$ and $\lambda_F$, on the filters
width and on efficiency of both channels.
$$P_{FW} = P_F\eta\frac{T_W}{T_F} \tag{12}$$
The signal measured in the water vapor channel, $\tilde{P}_W$, is the addition of both water vapor
backscatter, $P_W$, and the fluorescence backscatter, $P_{FW}$,
$$\tilde{P}_W = P_W + P_{FW} = P_W + P_F\eta\frac{T_W}{T_F} \tag{13}$$
One should remember, that the fluorescence spectrum, even for the same type of aerosol, can vary
with altitude and from observation to observation, which influences $\eta$. To minimize this influence
it is desirable to keep $\lambda_W$ and $\lambda_F$ as close as possible.





If the signals in the water vapor Raman and fluorescence channels are separated into co-
polarized (*II*) and cross-polarized ($\perp$) components, in respect to laser polarization state, their
powers in water vapor Raman channel are given respectively by Eq. 14 and Eq. 15
$$\tilde{P}_W^{\parallel} = P_W^{\parallel} + P_F^{\parallel}\eta\frac{T_W}{T_F} \tag{14}$$

$$\tilde{P}_W^{\perp} = P_W^{\perp} + P_F^{\perp}\eta\frac{T_W}{T_F} = \delta_W P_W^{\parallel} + \delta_F P_F^{\parallel}\eta\frac{T_W}{T_F} \tag{15}$$

Here $\delta_F$ and $\delta_W$ are the fluorescence and water vapor Raman depolarization ratios, defined in Eq.16
$$\delta_F = \frac{P_F^{\perp}}{P_F^{\parallel}} \qquad \delta_W = \frac{P_W^{\perp}}{P_W^{\parallel}} \tag{16}$$

Here we assume that depolarization ratio of fluorescence is the same at the wavelengths $\lambda_W$ and
$\lambda_F$. This assumption is usually valid, because emission is normally from the lowest singlet state,
so the depolarization ratio is spectrally independent (Lakowicz, 2006).
Due to the presence of fluorescence, the depolarization ratio measured in the water vapor
Raman channel is given by Eq.17
$$\tilde{\delta}_W = \frac{\tilde{P}_W^{\perp}}{\tilde{P}_W^{\parallel}} = \frac{\delta_W P_W^{\parallel} + \delta_F P_F^{\parallel}\eta\frac{T_W}{T_F}}{P_W^{\parallel} + P_F^{\parallel}\eta\frac{T_W}{T_F}} \tag{17}$$

Here $\delta_W$ is the depolarization ratio that would be measured in the water vapor Raman channel in
the absence of atmospheric fluorescence. From Eq.9, 10, 14, 15, 17 parameter $\eta$ can be derived
from lidar measurements, such as water vapor mixing ratio, $\tilde{n}_V$, depolarization ratio $\tilde{\delta}_W$ and
fluorescence backscattering $\beta_F$ :
$$\eta = \frac{\tilde{n}_W}{\beta_F}\frac{K_F}{K_W}n_R\frac{(1+\delta_F)(\tilde{\delta}_W - \delta_W)}{(1+\tilde{\delta}_W)(\delta_F - \delta_W)} \tag{18}$$

It should be note, that the choice of calibration constants $K_F$, $K_W$ does not influence $\eta$, because $\tilde{n}_W$
and $\beta_F$ are calculated with the same calibration constants. Finally, the increase of WVMR $\Delta n_W$
induced by the fluorescence can be calculated following Eq.19:





$$\Delta n_W = K_W \frac{P_F \eta \frac{T_W}{T_F}}{P_R} \frac{T_R}{T_W} = \frac{K_W}{K_F} \eta \beta_F \frac{1}{n_R}$$
(19)

As soon as parameter $\eta$ is found from Eq.18, we can estimate error $\Delta n_W$ from $\beta_F$, which in
case of LILAS is measured at 466 nm. In such estimation we have to assume that relationship
between fluorescence at 466 nm and 408 nm is constant (independent of height). Possibility to
perform correction from single – channel fluorescence measurements was discussed by Reichardt
et al. (2023), where it was shown, that for 466/408 nm channels, correction actually may depend
on height. Corresponding analysis based on our measurements will be presented in Sect.3.2.
We should mention, that when depolarization of water vapor Raman is available, the
contribution of fluorescence to WVMR can be obtained without using $\eta$. From Eq.18 and Eq.19 it
comes:
$$\Delta n_W = \tilde{n}_W \frac{(1+\delta_F)\left(\tilde{\delta}_W - \delta_W\right)}{(1+\tilde{\delta}_W)\left(\delta_F - \delta_W\right)}$$
(20)

However, such correction can be performed only at low altitudes, where signal-to-noise ratio in
cross-polarized water vapor channel is high enough.

**3. Experimental results**
In May – June 2023, the Canadian forest fires were at the origin of numerous smoke layers
observations in a wide range of altitude, ranging from the BL to the tropopause. The Boreal
wildfire season in 2023 started anomalously early. A wildfire in Alberta, Canada at 53.2° N, 115.7°
W has produced an intense Pyrocumulonimbus (PyroCb) cloud on 5 May with the minimum
satellite-derived infrared brightness temperature of -66° C, which should correspond to 10-11 km
altitude according to the local radiosoundings. In order to describe the long-range transport of the
smoke plume produced by this event, we use UV absorbing Aerosol Index (AI) measurements by
the Ozone Monitoring and Profiling Suite (OMPS) Nadir Mapper (NM) instrument onboard Suomi
NPP satellite mission (Flynn et al., 2014). AI is widely used as a proxy of the amount of absorbing
aerosols (e.g smoke, dust, ash) and its dimensionless value is proportional to the altitude of the
aerosol layers. The AI values above 15 are usually associated with the smoke plumes at or above
the tropopause (Peterson et al., 2018 and references therein), whereas the maximum AI value
reported by OMPS-NM instrument for the Alberta event amounted to 19.9.





Fig. 2 displays the spatiotemporal evolution of the smoke plume from the Alberta event
represented by the areas of enhanced AI observed between 5 – 21 May. The smoke in the upper
troposphere and lower stratosphere (UTLS) is carried by the westerly winds, crossing the Atlantic
in about 1 week before reaching Moscow region by 15 May. On that date, the Moscow lidar has
detected the smoke layer at 10-11 km (see Sect. 2). The plume was then further advected across
Eurasia towards northeastern Siberia. By 22 May the smoke plume completes its first
circumnavigation (not shown) and passes over Lille on 23 May and then over Moscow for the
second time around 27 May. Thus, we can expect, that the smoke layers observed over Lille and
Moscow have the same source.

### *3.1 Variability of fluorescence depolarization ratio*
At the first stage of our research we focused on the variability of the fluorescence
depolarization ratio with aerosol types. The main attention was paid to smoke particles, because
they provide the strongest impact on the Raman water vapor measurements due to their high
fluorescence capacity.
Spatio-temporal distributions of the aerosol elastic and fluorescence backscattering
coefficients ($\beta_{532}$ and $\beta_F$), on the night 26-27 June 2023, are shown in Fig.3. Dense smoke layer
with $\beta_F$ as high as $7.0 \times 10^{-4}$ Mm$^{-1}$sr$^{-1}$ occurs within the 4.0 -10.0 km height range. The relative
humidity increases from 40% at 4 km to RH>90% at 7 km where formation of ice crystals starts.
Vertical profiles of aerosol elastic and fluorescence backscattering coefficients ($\beta_{532}$ and $\beta_F$),
together with fluorescence capacity, are shown in Fig.3c. Inside the smoke layer, $G_F$ is about $3 \times 10^{-4}$,
which is a typical value for smoke whereas, above 6 km, it decreases due to ice formation. Ice
crystals increase the particle depolarization ratio $\delta_{532}$ from 3% at 6 km to 20% at 8 km.
Fluorescence signals are strongly depolarized. Inside the BL, $\delta_F$ is about 60% whereas above 2 km
it drops to approximately 45%. The processes of hygroscopic growth and ice formation do not
provide a noticeable impact on $\delta_F$ value. During May – June observations, the depolarization ratio
of smoke varied mainly inside the 45-55% range.
As discussed in our previous publications (Veselovskii, et al., 2022; Hu et al., 2022), the
fluorescence capacity of aged smoke varies inside the $(2.5-5.5) \times 10^{-4}$ range, probably due to the
changes in smoke composition and conditions of atmospheric transport. However, during Alberta
fires, several smoke plumes with high $G_F$ have been observed. The highest fluorescence capacity





was observed on the night 16-17 June 2023. Vertical profiles of aerosol properties for this episode
are shown in Fig.4. Dense smoke layers with fluorescence backscattering exceeding $10.0 \times 10^{-4}$
$Mm^{-1}sr^{-1}$ occurred within 7.0 -9.0 km height range. In this case, the maximal value of fluorescence
capacity reached $10.0 \times 10^{-4}$. Fluorescence depolarization ratio is about 50% through the entire
smoke layer and the process of ice formation (just like in Fig.3d) does not influence $\delta_F$. Thus, in
May - June 2023 strong variations of $G_F$ in the $(2.5-10.0) \times 10^{-4}$ range were accompanied by
relatively small variations of $\delta_F$ remaining in the 45 - 55% interval.
It is known that in the UTLS smoke particles can reach depolarization ratio, $\delta_{532}$ , as high as
15-20% (Burton et al., 2015; Haarig et al., 2018; Hu et al., 2019; Ohneiser et al., 2020). High
values of the depolarization ratio are usually attributed to the complex internal structure of smoke
particles (Mishchenko et al., 2016). Two smoke events in the UTLS, characterized by enhanced
$\delta_{532}$, on 28-29 May and 3-4 June 2023, are illustrated on Fig.5. On 28-29 May, three smoke layers,
at ~ 3.5, 6.5 and 11.5 km can be distinguished. High depolarization ratios, reaching 40% at altitudes
of 9.8-10.5 km, are due to ice clouds. In the lower smoke plumes ranging between 3.5 and 6.5 km,
the particle depolarization did not exceed 8% whereas above 11 km $\delta_{532}$ increases to 15%. High
values of $\delta_{532}$ observed in the UTLS correlate with increase of $G_F$ and with fluorescence
depolarization, $\delta_F$, up to $7.0 \times 10^{-4}$ and 70% respectively. Similar behavior was observed on 3-4
June, where depolarization ratio, $\delta_{532}$, above 11.5 km increased up to 15%, simultaneously with an
increase of $G_F$ and $\delta_F$ up to $9.5 \times 10^{-4}$ and 70% respectively. Thus, change in particle morphology
may affect the depolarization ratio of fluorescence. Another possibility is that, in the UTLS, not
only the particle structure can change, but composition as well. At the current stage of analysis,
we are not yet able to conclude about the mechanisms explaining the increase of fluorescence
depolarization in the UTLS.
We did not observe the effect of atmospheric humidity on smoke fluorescence
depolarization. However, inside the BL the observed hygroscopic growth was accompanied by an
increase of $\delta_F$. During the 9-16 June 2023 period numerous particle hygroscopic growth cases were
observed in the BL. One of such cases, on the night of 12-13 June, is shown in Fig.6. The relative
humidity increases inside the BL from 50% to 70% causing an increase of $\beta_{532}$ near the BL top.
Depolarization ratio $\delta_{532}$ decreases with height, since the particles in the process of hygroscopic
growth become more spherical. The fluorescence depolarization ratio, however, increases inside
the boundary layer from 50 to 70%.



All results obtained during 9-16 June, showing dependence of $\delta_F$ and $\delta_{532}$ on the relative
humidity, are summarized in Fig.7. Particle depolarization $\delta_{532}$ systematically decreased with RH
but, on 16 June, this dependence is not monotonic which could be due to the change of aerosol
composition with height.  At low RH (below 30%), fluorescence depolarization ratio is about 50%.
However, at RH about 90%, $\delta_F$ increases up to 70%. One possible explanation of $\delta_F$ behavior can
be an increase of rotational mobility of the molecules in the process of particle water uptake.

***3.2 Fluorescence spectrum sampled with a with 5-channel lidar***
The results presented in the previous section were obtained with a single channel
fluorescence lidar. However, for analyzing the variability of smoke properties (for example,
increase of fluorescence capacity with height) it is important to have information about total
fluorescence spectrum. Moreover, to estimate the fluorescence contamination in the Raman water
vapor channel, a relationship between fluorescence backscattering at 466 nm and 408 nm is used.
Thus we need to know the variability of the fluorescence spectrum in the short wavelength region.
In our recent work (Veselovskii et al., 2023) we presented the first results obtained with a 5-
channel fluorescence lidar in operation at the GPI. This lidar is able to measure fluorescence
backscattering in 5 spectral intervals centered at 438, 472, 513, 560, and 614 nm. In May – June
2023, several smoke plumes originating from Alberta fires were transported over Moscow.
Although Lille and Moscow are very distant from each other, smoke plumes observed have the
same origin, hence the fluorescence spectra measured over Moscow are quite helpful for the
analysis of Lille data.
Fig.8 presents fluorescence spectral backscattering coefficients, $B_\lambda$, for 3 smoke events
detected in the UTLS above 10, 8 and 10 km for 15, 31 May and 20 June 2023, respectively. On
15 and 31 May smoke layers are also present inside the 4-6 km range. Inside the BL the strongest
fluorescence is systematically detected in the 438 nm channel while, at higher altitudes, the
maxima shifts to 560 nm. As follows from Fig.8d-f, the ratio $B_{560}/B_{438}$ remains in the range 0.4 -
0.7 inside the BL whereas this ratio increases above 2.0 in the UTLS. Thus, for smoke events the
maxima of the fluorescence spectrum shifts with height towards longer wavelengths. The ratio
$B_{513}/\beta_{355}$ also increases with height and, above 10 km, it reaches the values of $1\times10^{-5}$ nm$^{-1}$. In the
UTLS, the maximal fluorescence capacity, $G_F$, measured by LILAS at 466 nm (with 44 nm
bandwidth filter) was about $10\times10^{-4}$. In the smoke layer, the ratio of backscattering coefficients





$\beta_{355}/\beta_{532}$ is about 2, so the maximal ratio $B_{466}/\beta_{355}$ derived from LILAS measurements is about
$1.1\times10^{-5}$ nm$^{-1}$. Thus, values obtained over Lille and over Moscow are in agreement.

The fluorescence spectra obtained for the above mentioned smoke plumes are shown in

Fig.9. The values of $B_\lambda$ are normalized to $B_{438}$. Inside the BL, the maximum of fluorescence is
measured at 438 nm and it decreases with wavelength. In the smoke layers within 4 - 6 km, the
maximum of fluorescence is observed at 513 nm while, in the UTLS, the maximum shifts to 560
nm.

When applying Eq.19 to estimate the contribution of smoke fluorescence into the Raman

water vapor channel of LILAS, we assume that the ratio of the fluorescence backscattering at 466
nm to 408 nm is constant. For the lidar in operation at GPI, the shortest available wavelength is
438 nm, therefore, at least, one can estimate the variability of the ratio $B_{472}/B_{438}$. Fig. 10 presents
vertical profiles of $B_{472}/B_{438}$ for 11 smoke events occurring during the 15 May – 20 June 2023
period. Inside the BL, this ratio varies in the 0.6 – 1.0 range. Lowest values correspond to urban
aerosols while, values of $B_{472}/B_{438}$ close to 1.0, probably indicate the presence of smoke particles
inside the BL. Smoke layers start mainly above 4.0 km and $B_{472}/B_{438}$ shows a tendency to increase
in the UT. It is interesting that, for the period 15 May – 1 June, the ratio was close to 1.5 whereas
after 1 June, it became close to 1.0, which can be related to changing of smoke source. Mean value
of $B_{472}/B_{438}$ in the 4.0 – 11.0 km range over all observations is 1.38 with standard deviation of 0.23
(relative variation is about 17%). The wavelength separation between 466 nm and 408 nm channels
is 1.7 larger, so one can expect variation of $B_{466}/B_{408}$ in the smoke layer up to ~30%. This is a very
rough estimation, but it points out the difficulties to face when the estimation of the fluorescence
contamination to the Raman water vapor channel is performed from a single fluorescence channel
at 466 nm. This issue was also discussed in the publication of Reichardt et al. (2023).

***3.3 Estimation of fluorescence impact on water vapor Raman measurements***

Measuring the depolarization ratio in the water vapor Raman channel provides an

opportunity to control/evaluate the presence of fluorescence leak in this channel. These
depolarization measurements were performed in Lille during May – June 2023. Vertical profiles
of water vapor depolarization ratio $\tilde{\delta}_W$ together with $\tilde{n}_W$, $\beta_{532}$, $\beta_F$, and $G_F$ are shown in Fig.11 for
the night 8-9 and 10-11 June 2023. On 8-9 June aerosols are confined mainly below 5 km. The
fluorescence capacity is about $1.0\times10^{-4}$ below 3.0 km, but above, $G_F$ increases up to $2.5\times10^{-4}$,





indicating to the presence of smoke. The depolarization ratio in the water vapor channel is about
2% in the height range 1.5 km – 3.5 km. The values of $\tilde{\delta}_W$ ranging inside 1.8%÷2.0% were
observed for this height range, where contribution of fluorescence was insignificant.
Depolarization ratio $\delta_W$ is low, because the interference filter in water vapor channel selects only
strongest Q-branch lines and most of rotational lines are blocked. Contribution of fluorescence
becomes noticeable above 3.5 km where $n_W$ drops, resulting in an increase of $\tilde{\delta}_W$ up to ~3%. There
is also increase of $\tilde{\delta}_W$ up to 2.2% below 1.0 km, where fluorescence backscattering is enhanced.
Similar values of $\tilde{\delta}_W$ were observed on 10-11 June, where depolarization ratio increases up to
2.5% inside the smoke layer observed at ~3.75 km and below 2.0 km.

As discussed in section 2.2, the contribution of fluorescence to the WVMR can be derived

from Eq.20 if $\tilde{\delta}_W$ and $\delta_F$ are measured. Fig.12 presents the modeling of the relative error $\frac{\Delta n_W}{\tilde{n}_W}$,
introduced by the fluorescence to WVMR as a function of $\tilde{\delta}_W$. Computations are performed for
fluorescence depolarization ratio $\delta_F$=50%, 60%, 70% to include both smoke and urban particles.
Depolarization ratio in the Raman water vapor channel in the absence of fluorescence was assumed
to be $\delta_W$=2%. For depolarization ratio $\tilde{\delta}_W$ below 3% the relative error $\frac{\Delta n_W}{\tilde{n}_W}$ did not exceed 3%.
As follows from the fluorescence spectra in Fig.9, the fluorescence of urban particles increases
towards short wavelengths, thus one can expect impact of the urban aerosol fluorescence on vapor
measurement. In practices, however, we did not observe $\tilde{\delta}_W$ exceeding 3% in the BL thus,
contribution of aerosol in the BL is not critical. The reason is due to the low fluorescence capacity
(about one order lower than that of smoke) and higher water vapor content, comparing to free
troposphere.

Profiles of $\tilde{\delta}_W$ shown in Fig.11 become noisy at heights where $n_W$ is low, and $\tilde{\delta}_W$ can not be

used for correction of fluorescence effect in the UT. To overcome this, one derived the parameter
$\eta$ from Eq.18 at low altitudes where $\tilde{\delta}_W$ is available, and, then, this $\eta$ is used to calculate $\Delta n_W$ from
Eq.19 in the entire height range. In such an approach, however, one has to assume that relationship
between fluorescence cross sections at 466 nm and 408 nm remains constant with height. As





discussed in previous section, such assumption can yield significant bias in calculation of $\Delta n_W$,
and, at this stage, we do not provide corrected profiles of WVMR.

For accurate calculation of $\eta$ one needs smoke events with strongly enhanced $\tilde{\delta}_W$, which is

usually observed in the dry smoke layers. Such suitable events are shown for the night 26-27 May
and 5-6 June 2023 in Fig.13. On 26-27 May a smoke layer characterized by high fluorescence ($\beta_F$
up to $5\times10^{-4}$ Mm$^{-1}$sr$^{-1}$) and low $\tilde{n}_W$ (below 0.2 g/kg) is observed at 3.5 km. Fluorescence
depolarization ratio is about 47% and $\tilde{\delta}_W$ increases from 2% up to 12% in the middle of this layer.
Parameter $\eta$ calculated from Eq.18 inside this smoke layer is about $2\times10^{-3}$ (g/kg)/(Mm$^{-1}$sr$^{-1}$). On
5-6 June the depolarization ratio $\tilde{\delta}_W$ in the smoke layer increased up to 10% and value of $\eta$ is very
similar. Parameters $\eta$ derived for several smoke episodes vary in the range $(2\div2.5)\times10^{-3}$
(g/kg)/(Mm$^{-1}$sr$^{-1}$). For the estimate of $\Delta n_W$ one used the mean value of $\eta=2.25\times10^{-3}$ (g/kg)/(Mm$^{-$
$^1$sr$^{-1}$), which is suitable only for smoke, while for particles in the BL, $\eta$ can be different. However,
in the BL, low depolarization ratio $\tilde{\delta}_W$ prevented us from calculating $\eta$.

Fig.14 presents vertical profiles of WVMR, fluorescence backscattering and error $\Delta n_W$

introduced by the fluorescence in WVMR on 26-27 May, 28-29 May and 16-17 June. Smoke layers
with strong fluorescence occurred systematically in our UT observations.  The current LILAS
system is not powerful enough for deriving accurate water vapor measurements above 10 km,
however increase of $\tilde{n}_W$ in fluorescent smoke layers is visible. We remind that Eq.19 for $\Delta n_W$
contains factor $\dfrac{1}{n_R}$ (inverse relative change of nitrogen number density), thus the fluorescence
impact on WVMR increases with height. The uncertainties $\dfrac{\Delta n_W}{\tilde{n}_W}$ for all events considered are
shown in Fig.14d. On 26-27 and 28-29 May the uncertainty at 11 km is of the order of 100%. On
16 June the smoke layer is lower (at 9 km) and the uncertainty is about 50%. Our demonstration
shows that smoke fluorescence can significantly impact water vapor measurements. The proposed
approach, based on the analysis of the depolarization ratio of the water vapor signal, has potential
for estimation of corresponding errors and their correction.

**4. Conclusion**



Modern fluorescence spectroscopy widely uses depolarization measurements of the
fluorescence emission induced by polarized laser radiation. However, in application to lidar
atmospheric observations, measurement of fluorescence depolarization ratio, presented in this
study, is one of the first efforts in this direction. Analysis of more than 30 spring and summer
smoke events allows evaluation of the main aerosol intensive properties, including fluorescence
capacity, particle and fluorescence depolarization ratio. The fluorescence capacity varied within
$(2.5 \div 10.0) \times 10^{-4}$ range while the particle depolarization ratio $\delta_{532}$ remained below 10%. However,
in spite of strong $G_F$ variation, $\delta_F$ was remaining within a relatively narrow interval 45-55%.
Additional observations revealed that for smoke plumes in the upper troposphere $\delta_{532}$ increased up
to 15% and fluorescence depolarization increased up to 70%. At the moment, one cannot fully
explain the mechanism responsible for this $\delta_F$ increase. It can be related to complex particle
internal structure at high altitudes, as well as to the change of the chemical composition, revealed
by the shift of the maximum of fluorescence spectra to longer wavelengths in the upper troposphere
(Fig. 9).
Inside the BL, the fluorescence depolarization ratio was higher than that of smoke and varied
inside the 50-70% range. Moreover, fluorescence depolarization ratio of urban particles strongly
depends on the relative humidity and, in contrast to elastic scattering, depolarization of
fluorescence increases with RH. One possible origin of this phenomena could be attributed to an
increase of rotational mobility of the molecules involved in the process of water uptake since $\delta_F$
increases when rotation time of molecules becomes comparable with time of fluorescence
emission.
The depolarization ratio of Raman water vapor backscatter, in the absence of fluorescence,
appears to be quite low ($\delta_W = 2 \pm 0.5\%$), because the narrowband interference filter in the water vapor
channel selects only strong Q-lines of the Raman spectrum. As a result, the depolarization ratio of
Raman water vapor backscatter is sensitive to the presence of strongly depolarized fluorescence
signals. Contribution of fluorescence to the WVMR can be calculated from $\tilde{\delta}_W$ with the only
assumption that $\delta_F$ remains constant within 408 - 466 nm. However, the depolarization ratio of
Raman water vapor backscatter is weak and measurements are only possible up to the middle
troposphere with the lidar used in the work, while the problem of fluorescence interference is the
most crucial in UTLS. To solve this problem, one can derive $\eta$, a parameter linking fluorescence



at 466 nm and at 408 nm. This parameter is calculated for specific smoke events, at low altitudes,
and then is used for processing all observations and altitudes.
Such an approach relies on the assumption that the ratio of fluorescence between 466 nm
and 408 nm remains constant and allows only a rough estimation of the correction term for the
water vapor mixing ratio, $\Delta n_W$. One possible solution to improve accuracy of $\Delta n_W$ is to implement
an additional and shorter wavelength channel (438 nm or even shorter). Another technical solution
could be considered as the depolarization ratio of Raman water vapor backscatter is low, therefore
the 408 nm component can be efficiently selected with a polarizing cube. The depolarized channel
then can be used for fluorescence measurements. Polarizing cube works in a wide spectral range,
so one can select the region outside of the water vapor spectrum (400 nm – 418 nm) for
fluorescence monitoring. We plan this experiment as well as other innovative approaches with our
future high power fluorescence lidar, LIFE (Laser Induced Fluorescence Explorer), whose start of
operation is scheduled at the beginning of 2024.

*Data availability*. Lidar measurements are available upon request
(philippe.goloub@univ-lille.fr).

*Author contributions*. IV processed the data and wrote the paper. QH and TP performed the
measurements in Lille. PG supervised the project and helped with paper preparation. WB modified
LILAS for polarization measurements. MK and NK performed the measurements in Moscow. SK
analyzed transport of smoke layers and RM derived RH profiles from lidar measurements.
.

*Competing interests*. The authors declare that they have no conflict of interests.

**Acknowledgement**
We acknowledge funding from the CaPPA project funded by the ANR through the PIA under
contract ANR-11-LABX-0005-01, the "Hauts de France" Regional Council (project ECRIN) and
the European Regional Development Fund (FEDER). ESA/QA4EO program is greatly
acknowledged for supporting the observation activity at LOA. The work from Q. Hu was
supported by Agence *Nationale* de Recherhce ANR (*ANR-21-ESRE-0013) through the*
OBS4CLIM project. Development of fluorescence lidar in Moscow was supported by Russian
Science Foundation (project 21-17-00114). The work of S. Khaykin was partly supported by the
Agence Nationale de la Recherche (ANR) 21-CE01- 335 0007-01 PyroStrat project.



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



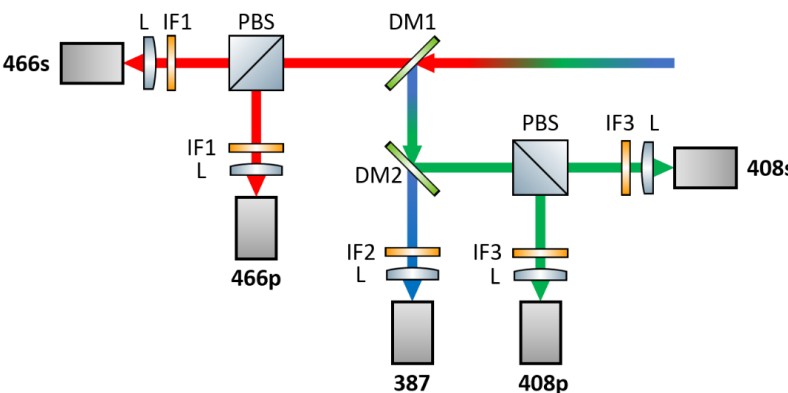


Fig.1. Optical layout of depolarization measurements at 408 nm and 466 nm wavelengths. L –
lens; IF1 - IF3 – interference filters, DM1, DM2 – dichroic mirrors, PBS – polarizing cube.


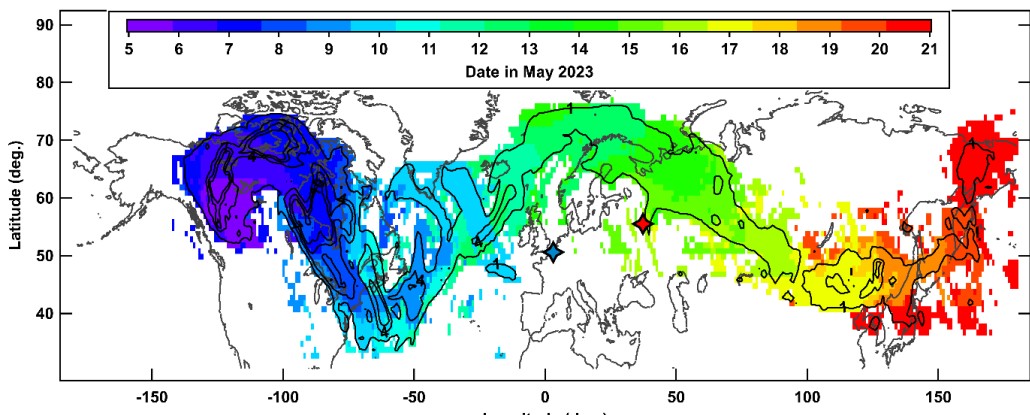


Fig.2. Spatiotemporal evolution of the smoke plume from the wildfire event in Alberta, Canada on
5 May 2023. Color-filled time-coded areas indicate the occurrences of Aerosol Index (AI) values
from OMPS-NPP instrument exceeding 0.5. The actual AI values (1-10) are shown in contours.
The blue and red-filled stars indicate the location of Lille and Moscow lidar stations respectively.

583



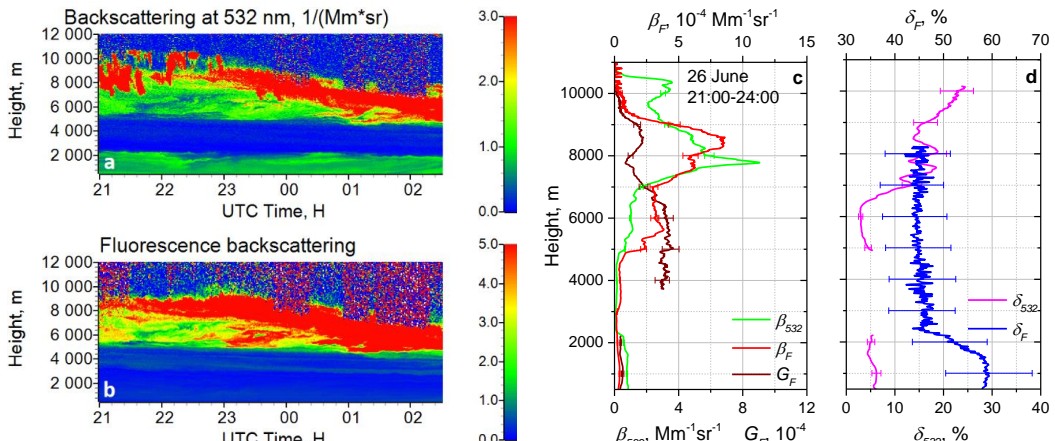

Fig.3. Smoke event on the night 26-27 June 2023. Spatio-temporal distributions of (a) aerosol backscattering coefficient $\beta_{532}$ and (b) fluorescence backscattering $\beta_F$ (in $10^{-4}$ Mm$^{-1}$sr$^{-1}$). Vertical profiles of (c) the aerosol $\beta_{532}$ and fluorescence $\beta_F$ backscattering coefficients, the fluorescence capacity $G_F$; (d) the particle $\delta_{532}$ and the fluorescence $\delta_F$ depolarization ratios.

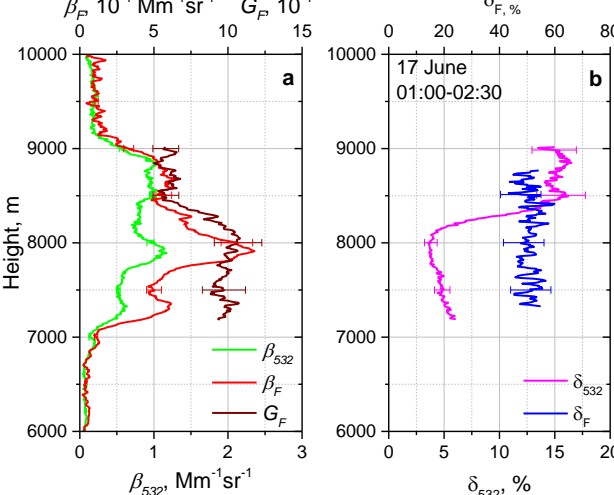

Fig.4. Vertical profiles of (a) aerosol $\beta_{532}$ and fluorescence $\beta_F$ backscattering coefficients, fluorescence capacity $G_F$ and (b) particle $\delta_{532}$ and fluorescence $\delta_F$ depolarization ratios on the night 16-17 June 2023 for period 01:00-02:30 UTC.






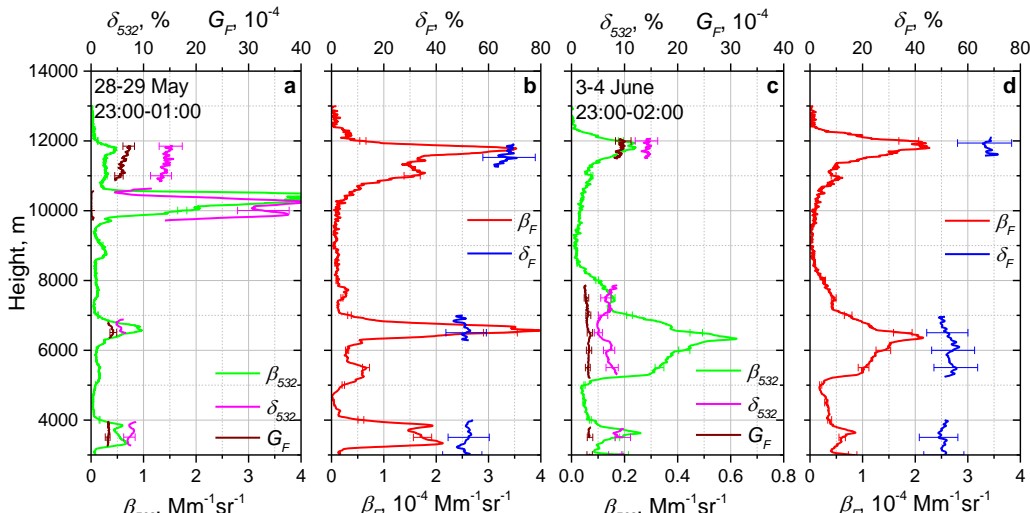

Fig.5. Vertical profiles of (a, c) backscattering coefficient $\beta_{532}$, particle depolarization ratio $\delta_{532}$,
fluorescence capacity $G_F$ and (b, d) fluorescence backscattering $\beta_F$ and fluorescence depolarization
ratio $\delta_F$ for two smoke episodes on the nights 28-29 May 2023 and 3-4 June 2023.




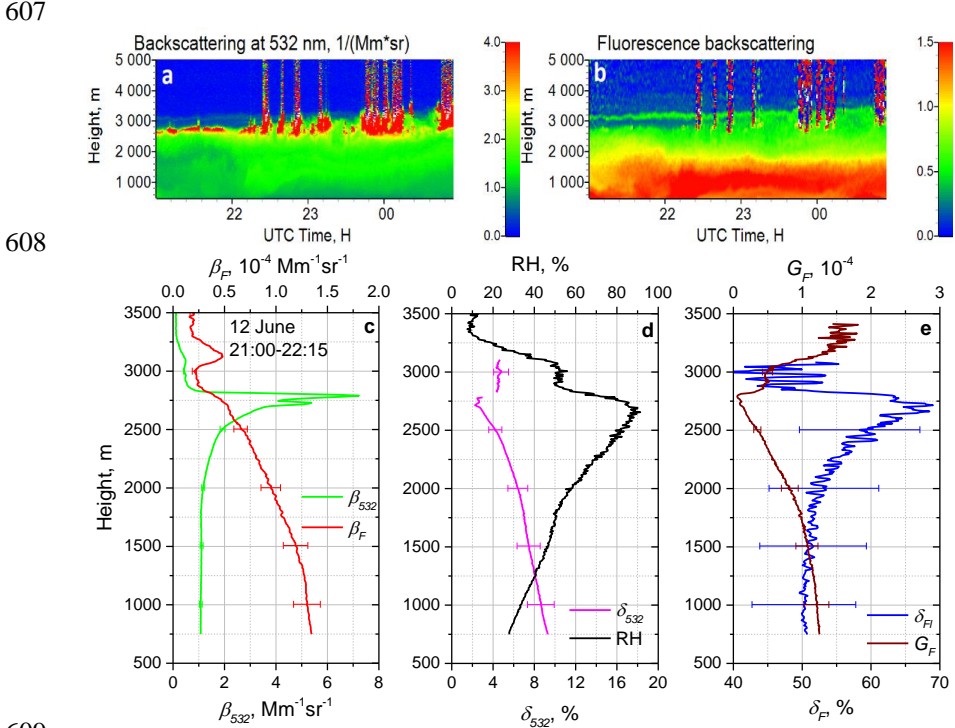


Fig.6. Particle hygroscopic growth in the boundary layer on the night 12-13 June 2023. Spatio-
temporal distributions of (a) aerosol backscattering coefficient $\beta_{532}$ and (b) fluorescence
backscattering $\beta_F$ (in $10^{-4}$ $Mm^{-1}sr^{-1}$). Vertical profiles of (c) aerosol $\beta_{532}$ and fluorescence $\beta_F$
backscattering coefficients; (d) particle depolarization ratio $\delta_{532}$ and the relative humidity RH; (e)
fluorescence depolarization ratio $\delta_F$ and fluorescence capacity $G_F$ for the time period 21:00-22:15
UTC.






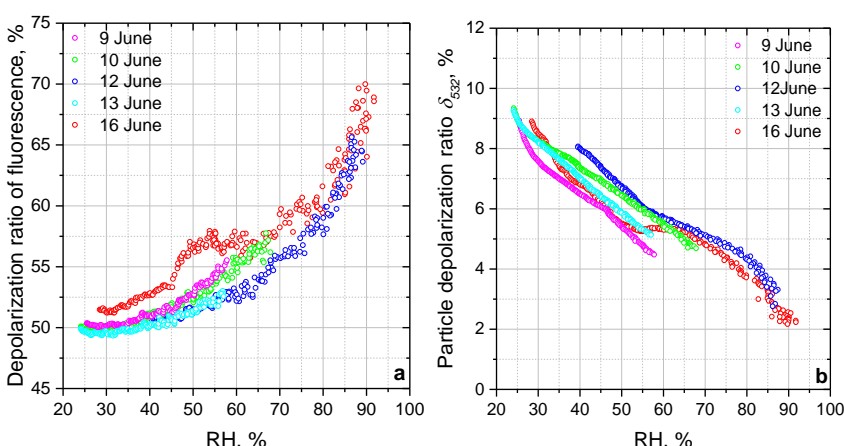

Fig.7. (a) Fluorescence depolarization ratio and (b) particle depolarization ratio $\delta_{532}$ as a function
of the relative humidity in the boundary layer for the measurements on 9, 10, 12, 13, 16 June 2023.

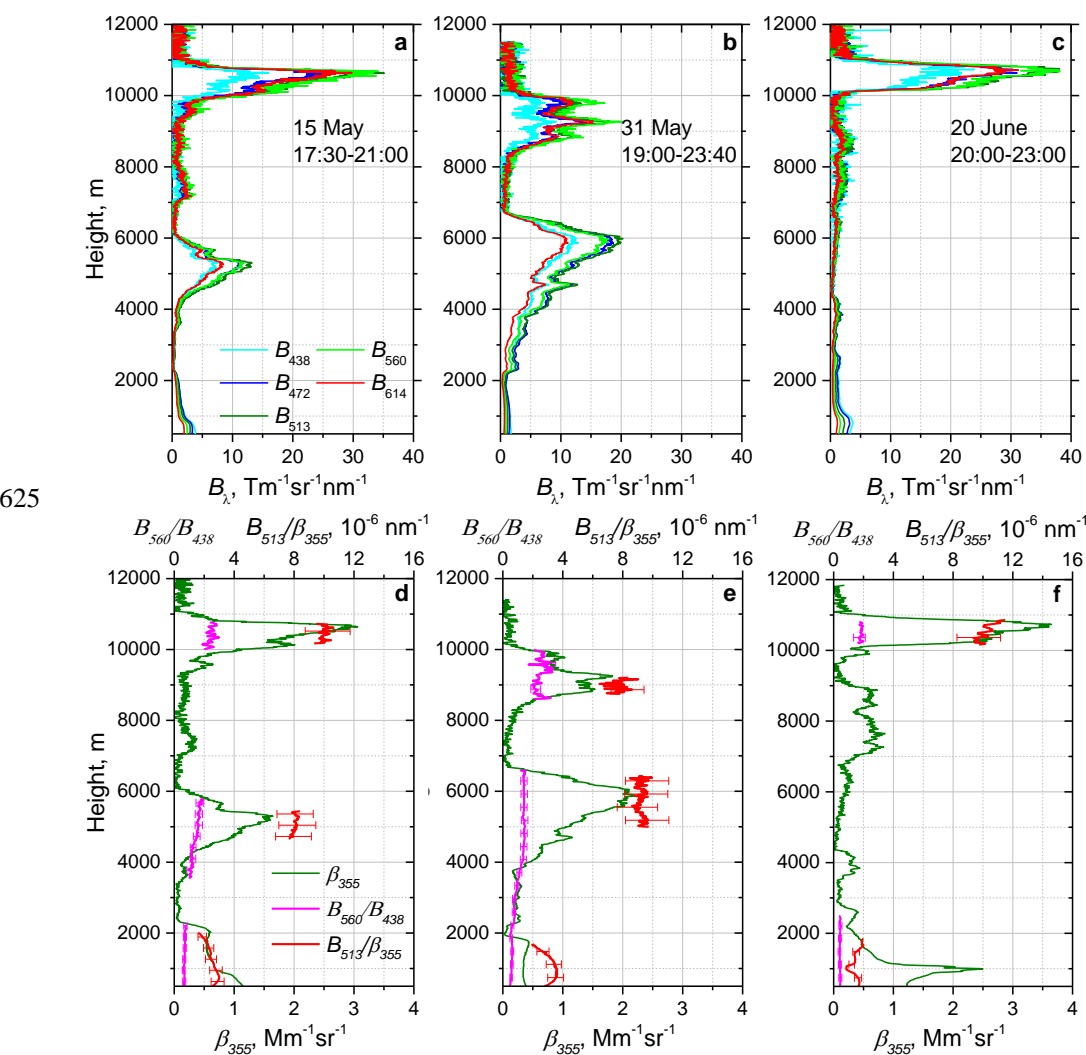


Fig. 8. Fluorescence measurements in Moscow on 15 May, 31 May, 20 June 2023. Vertical profiles of (a-c) fluorescence spectral backscattering coefficients $B_\lambda$ at 438, 472, 513, 560, 614 nm and (d-f) aerosol backscattering coefficient $\beta_{355}$, the ratio $B_{560}/B_{438}$ and $B_{513}/\beta_{355}$. Measurements were performed at an angle of 48 dg to horizon.

631





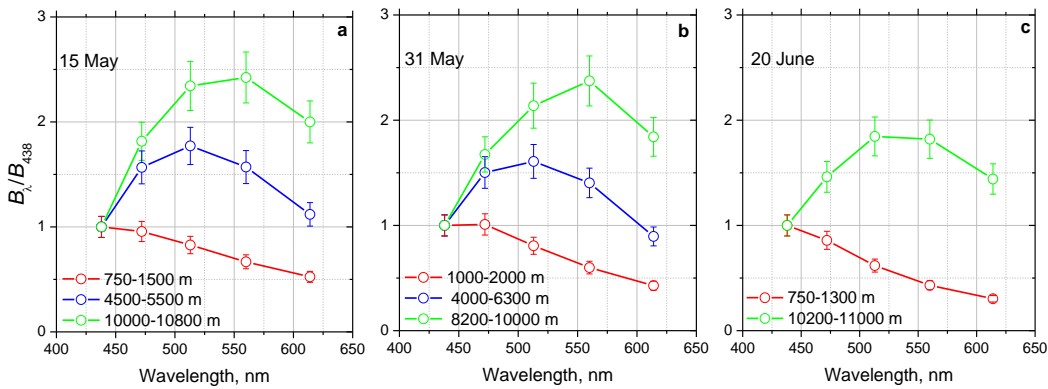

Fig.9. Fluorescence spectra $B_\lambda/B_{438}$ at different height intervals measured during smoke episodes on 15 May, 31 May, 20 June 2023 for the same temporal intervals as in Fig.8.





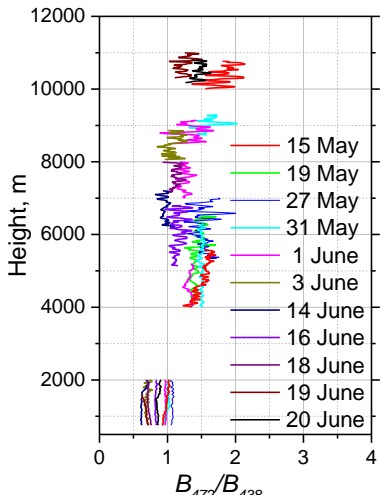

636

Fig.10. Height profiles of ratio $B_{472}/B_{438}$ for smoke episodes during 15 May – 20 June 2023. Smoke layers start above 4000 m.

639



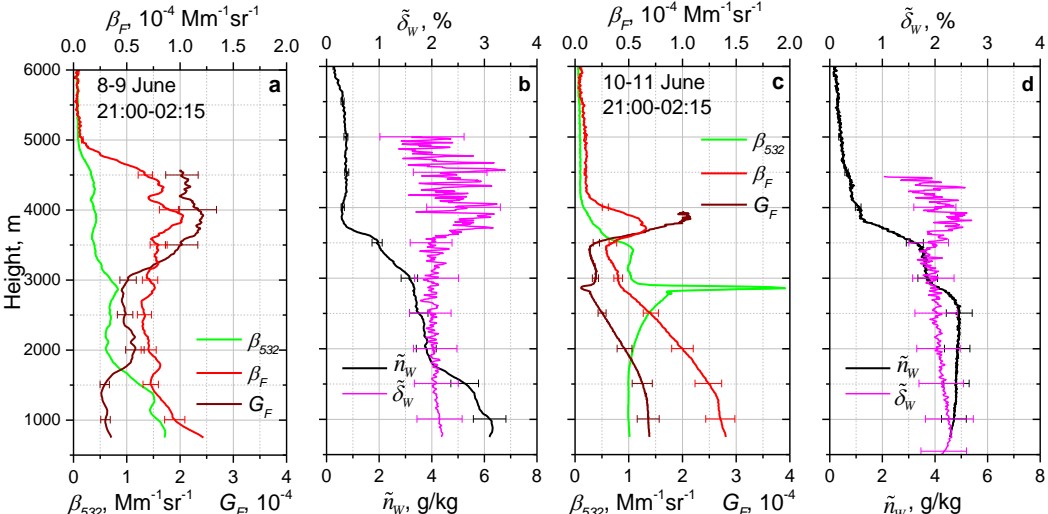

Fig.11. Impact of aerosol fluorescence on the depolarization ratio in the water vapor Raman channel on the nights 8-9 and 10-11 June 2023 at Lille. Vertical profiles of (a, c) particle backscattering $\beta_{532}$, fluorescence backscattering $\beta_F$, fluorescence capacity $G_F$ and (b, d) depolarization ratio $\tilde{\delta}_W$ of water vapor Raman signal and the water vapor mixing ratio $\tilde{n}_W$.





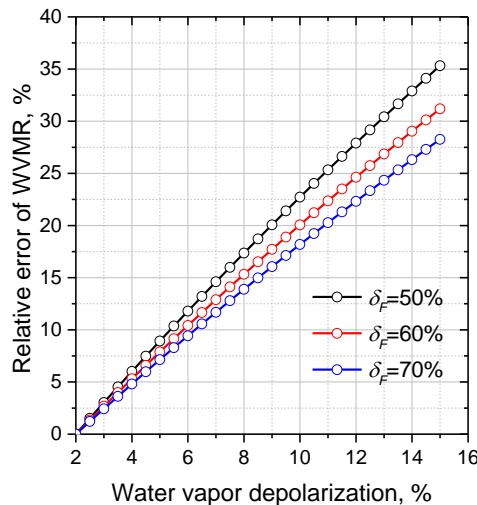

646

Fig.12. Relative error of water vapor mixing ratio (WVMR) $\dfrac{\Delta n_W}{\tilde{n}_W}$ induced by the fluorescence as

a function of depolarization ratio $\tilde{\delta}_W$ in the water vapor Raman channel for three values of
fluorescence depolarization ratio $\delta_F = 50\%$, $60\%$, $70\%$. The depolarization ratio of water vapor
Raman backscatter in the absence of fluorescence is assumed to be $\delta_W = 2\%$.



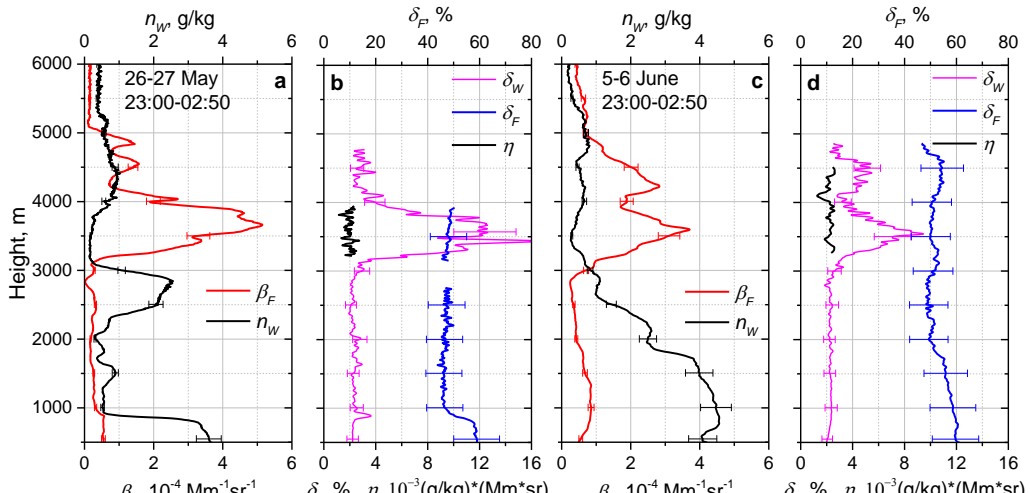


Fig. 13. Fluorescence measurements in Lille on the night 26-27 May and 5-6 June 2023. (a, c)
Vertical profiles of the fluorescence backscattering $\beta_F$, the water vapor mixing ratio $\tilde{n}_W$, (b, d) the
depolarization ratio of water vapor Raman signal $\tilde{\delta}_W$, the fluorescence depolarization ratio $\delta_F$ and
parameter $\eta$, describing contribution of fluorescence to the water vapor channel.







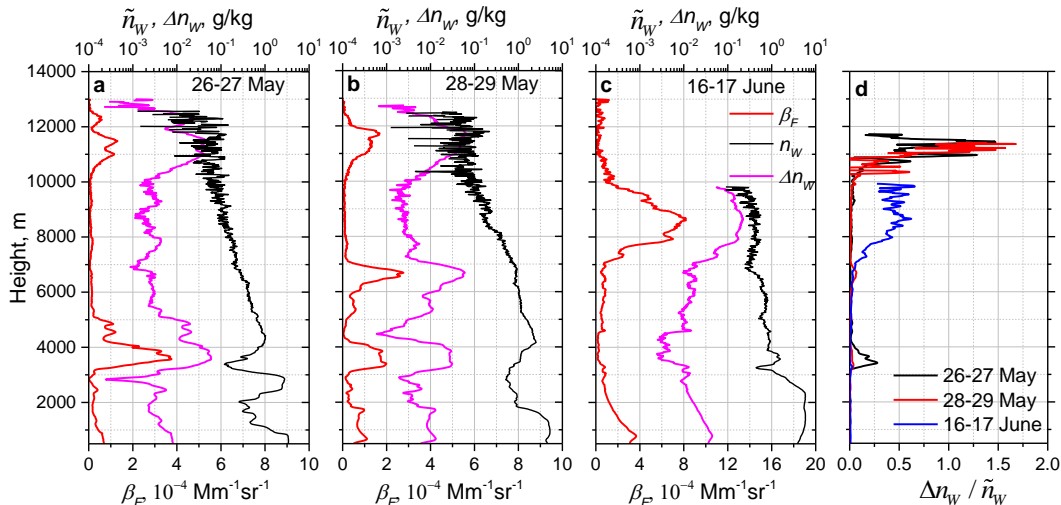

Fig.14. Impact of smoke fluorescence on the water vapor measurements. Vertical profiles of
fluorescence backscattering $\beta_F$, water vapor mixing ratio $\tilde{n}_W$ and bias in water vapor channel $\Delta n_W$
provided by the fluorescence of smoke for episodes on the nights (a) 26-27 May, (b) 28-29 May
and (c) 16-17 June 2023 for time interval 21:00-02:30 UTC. (d) Error $\dfrac{\Delta n_W}{\tilde{n}_W}$ introduced by smoke
fluorescence for three episodes.