# Peer review of "Derivation of depolarization ratios of aerosol fluorescence and water vapor Raman"

_Atmospheric Measurement Techniques, 2023_

## Referee Comment (RC1)

[revised manuscript text omitted]

 where $P_F^{\parallel}$ and $P_F^{\perp}$ are the powers of co- and cross-polarized fluorescence components. So in lidar measurements,  is the fluorescence depolarization ratio, $\delta_F$, is given as:

✱ The paper of Wang et al., 2023 refers to multi-wavelength lidar measurements using combined elastic-Raman-fluorescence data, the later using a 32-PMT lidar detector. I propose to omit this paper as it is not a pure single-channel fluo lidar.

$$\delta_F = \frac{P_F^{\perp}}{P_F^{\parallel}}.$$  (2)

*Therefore, the anisotropy is expressed as a function of the $\delta_F$ as follows:*

$$r = \frac{1-\delta_F}{1+2\delta_F}$$  (3)

For randomly oriented fluorophores with collinear absorption and emission dipoles, in the absence of rotational motion, the anisotropy  r=0.4 (Lakowicz, 2006), which corresponds to

$\delta_F$=33%. This is the minimal value one can expect in lidar measurements. Existence of  *any* angle between absorption and emission dipoles, as well as molecule rotation in the process of emission will increase $\delta_F$. Thus, measurement of fluorescence depolarization ratio may bring additional information about atmospheric aerosol. *as we will show below.*

→ *This is also written in line 82.*

Water vapor is a key atmospheric component playing essential role in the planet's radiative balance, and Raman lidars today are widely used for  *such* observations (Whiteman, 2003, Chouza et al., 2022 and references therein). However, when the UV laser beam passes through a smoke layer, the broadband fluorescence signal is induced and its spectrum includes the region of water vapor Raman lines. Thus, the signal in the water vapor channel (around 407.5 nm, when 354.7 nm *laser*

radiation is  *emitted* ation) becomes contaminated by the fluorescence backscatter signal (Immler et al.. 2005; Immler and Schrems, 2005). This contamination can be reduced by decreasing the width of the transmission band in the water vapor channel down to tenths of nm.

However, as it was shown recently, fluorescence still remains  *an* issue, especially inside the smoke layers in  *the upper free* troposphere (Chouza et al., 2022; Reichardt et al., 2023). *Reichardt et al., 2018*

*moreover,* epolarization measurements provide an opportunity to monitor the presence of fluorescence signal in Raman channel*s.* *the* Q-branch of water vapor Raman lines (near 407.5 nm)

provides a weakly depolarized backscatter, while fluorescence is strongly depolarized. Thus, the presence of fluorescence should increase the depolarization ratio of signal in the water vapor channel. Moreover, if *the* depolarization ratios of water vapor and fluorescence are known, the contribution of fluorescence to the measured water vapor mixing ratio (WVMR) can be evaluated.

In this article, *we* report and analyze, for the first time, *the* depolarization ratio of aerosol fluorescence and  of water vapor Raman backscatter from lidar observations performed at the ATOLL observatory (ATmospheric Observation at liLLe), Laboratoire d'Optique Atmosphérique, University of Lille, during dense smoke events  occurred  on May -

June 2023. We start with a description of the experimental setup in Sect.2.1 and derive, in Sect.

2.2, the main equations for estimating the fluorescence contribution to the water vapor Raman channel. In the first part of the results section (Sect.3.1), the fluorescence depolarization ratios over

ATOLL  are analyzed for different aerosol types. The measurements of fluorescence spectra performed with a new five-channel fluorescence lidar, operated in Moscow, are presented in

Sect.3.2. In Sect. 3.3, we analyze the depolarization ratio in the water vapor Raman channel and estimate the contamination of fluorescence to the derived WVMR.

profiles. Finally, in sect. 4 we present our conclusions.

**2. Experimental setup and data analysis**

**2.1 Lidar system**

In our study, two lidar systems  are considered. The first one, LILAS ((LIlle Lidar

AtmosphereS) is a multiwavelength Mie-Raman-Fluorescence lidar, whereas the second one is a multiwavelength fluorescence lidar operated by the General Physics Institute (GPI), Moscow (Veselovskii et al., 2023). Both systems are based on a tripled Nd:YAG laser (Q-Smart 450) with a 20 Hz repetition rate and pulse energy about 100 mJ at 355 nm. The Backscattered laser light in both systems is collected by a 40 cm aperture telescope and the lidar signals are digitized with (Licel)  transient recorders with 7.5 m range resolution, allowing simultaneous detection in the analog and photon counting mode.

LILAS allows the so called $3\beta+2\alpha+3\delta$ configuration, including three particle backscattering ($\beta_{355}$, $\beta_{532}$, $\beta_{1064}$), two extinction ($\alpha_{355}$, $\alpha_{532}$) coefficients along with three particle depolarization ratios. ($\delta_{355}$, $\delta_{532}$, $\delta_{1064}$) The Raman channel with $\alpha$ 407.54/0.3 nm spectral width interference filter allows also water vapor profiling. At the end of 2019, the lidar was modified to enable fluorescence measurements. A

part of the fluorescence spectrum is selected by a wideband interference filter of 44 nm width centered at 466 nm (Veselovskii et al. 2020).

In the fluorescence lidar of GPI only 355 nm wavelength is emitted, while fluorescence is measured in five spectral spectral intervals. The central wavelengths and widths of transmission bands are: (in parentheses)

438(29), 472(32), 513(29), 560(40) and 614(54) nm (Veselovskii et al., 2023). Thus, the fluorescence spectrum could be sampled. at five different wavelengths. ⊛ At GPI, the measurements were performed at an angle of 48 deg to the horizon. The strong sunlight background restricts the fluorescence observations of both systems to only the nighttime hours.

⊛ I would ask the authors to provide, in the Supplement Section a figure showing the transmission spectra of the 5-wavelengths mentioned, so that the readers can see if there are overlaping transmission + curves between the various filters. If any they the authors should refer to any induced errors in the detected lidar signals.

[revised manuscript text omitted]

over ... define city/region.

---

## Author Comment (AC1)

Response to Referee#1

We would like to thank the Referee for very detailed reading of the manuscript and numerous suggestions. In the process of revision we tried to follow his recommendations.

"It is a notable contribution to extending existing fluorescence and water vapor Raman lidar techniques and an interesting topic for the lidar community. The context and study goal are clear, and the results don't show major errors and omissions. However, there are many presentation styles, grammar, and typo errors. Is this a haste-writing? Therefore, I recommend it to be considered for publication after some revisions."

In the revised manuscript numerous corrections of style were introduced.

"In general, the structure of this manuscript needs to be improved. Some part of the abstract is read like a conclusion, such as line 31-38, while the conclusion is too detailed about the results discussion without extracting the main points. The introduction also can be more converging to the aims of this study, and a better context is needed between paragraphs, e.g. line 71-72, it seems water vapor just come out without connection with the above."

In revised manuscript we strongly decreased Abstract and Conclusion following Referee recommendations.

"For the scientific part, the typical values of fluorescence depolarization ratio (line 277-280), the hygroscopic growth cases in BL(line 307), and the assumption of "change of aerosol composition" (line 315) as well as "change in particle morphology may affect the depolarization ratio of fluorescence" "at RH about 90%, δF increases up to 70%." (means more spherical higher δF?), these above probably should be discussed, because it seems the results or shreds of evidence do not support them well."

We agree with Referee, that the additional supported evidences of observed effects are desirable. However, it is not always possible. For example, observed variations of fluorescence capacity should be related to variation of the smoke composition. However, to support of this statement, the independent in situ smoke sampling inside the smoke layer is needed, which was not available at the time. Some effects are observed in this study for the first time, and we are not able to provide the complete scientific explanation. For example, increase of $\delta_F$ with RH inside the PBL. It occurs, definitely, not due to the spherical shape, but, can be due to increase of molecules mobility inside the particle. Unfortunately, we have no information about rotational recombination time or about fluorescence life time of molecules in atmospheric aerosol, to provide quantitative analysis. Suggested mechanism is just one of possible, and additional studies are needed. Still, we think that it is important to present the results to community, it may stimulate others for research in this field.

Details:

"The title "**Derivation of aerosol fluorescence and water vapor Raman  depolarization ratios…**" is easy to mis-understand, "**Derivation of depolarization ratios of aerosol fluorescence and water vapor Raman backscatters…**" might be better."

We agree, the title is modified

Abstract:

"Line 16: "The total scattered power of" means to "the total backscattered power of"?"

Yes, corrected

"Line 19: regarding the same source of smoke, where is this measurement taken place relative to Moscow, so it is at Lille?"

Yes, at Lille. We had two systems, at Moscow and at Lille

"Line 27: "…fluorescence at 513 nm while, in the upper…" to "fluorescence at 513 nm, while in the upper""

Corrected

"Line 36: should be "the strongest…"""

Corrected

Line 38-40: suggest to re-write this sentence since it is difficult to understand.

Corrected. Now it is "As a result, depolarization ratio at the water vapor Raman is sensitive to the presence of strongly depolarized fluorescence backscattering and can be used for evaluation of corresponding uncertainty of the water vapor mixing ratio (WVMR) measurements."

Introduction:

"Line 57: "In fluorescence spectroscopy.." a comma is missing. And what is the motivation to introduce the anisotropy?"

Corrected. We modified the sentence, now it is "In the fluorescence spectroscopy, the polarization state of emission is described by the anisotropy (Lakowicz, 2006), introduced as:"

"Line 65-67: check the grammar."

Checked

"Line 67: "minimal" should be "maximal"?"

No, this is minimal. Observed depolarization should be higher than 33%.

"Line 68: it is unclear. Why the molecule rotation could increase the depolarization?"

If rotation is faster than the fluorescence life-time, the emission is completely depolarized. This is widely used technique for viscosity measurements, so we just provide reference to Lacowicz book.

"Line 80: Could the authors please be more specific about this issue "fluorescence still remains the issue"?"

Changed for "fluorescence still remains the source of uncertainties, especially when the water vapor mixing ratio (WVMR) is measured inside the smoke layers in the upper troposphere"

"Line 84: "weakly" to "weak""

Hm…, looks like weakly is correct…

"Line 149: "For both channels" a comma missing"

Corrected

Results :

"Line 293: "High depolarization ratios" means particle depolarization ratios? at 532 nm?"

Yes, we introduced it in the beginning of section 2.

"Line 295: "particle depolarization" to "particle depolarization ratio"?"

Yes, corrected

"Line 355: The lowest"

Corrected

"Line 358: please give a definition of the "ratio""

Done

"Line 360: please explain how is the standard deviation obtained."

For calculation of standard deviation we used the expression

$$s = \sqrt{\frac{1}{N-1}\sum_{i=1}^{N}(x_i - \bar{x})^2}$$

Where N is number of measurements $x_i$ and $\bar{x}$ is the mean value. This is standard definition and we are not sure, that it should be presented it in the manuscript.

"Line 406: the depolarization ratio of water vapor is due to the mixing with fluorescence?"

Yes

"Line 410-411: please check the grammar and re-write the sentence."

We corrected the sentence

"Line 251, 312, 354, 355, 375, and so on…: "5 – 21 May" and "9-16 June" Please unify the format everywhere, and also the symbols for indicating the parameters. "1.8%÷2.0%" to "1.8%-2.0%"? "On 26-27 and 28-29 May the uncertainty" There are a lot of such format or typo issues, I will not point out one by one."

In revised manuscript we use everywhere the AMT format "5-21 May", "5-10%"…

Conclusion :

"Line 426-427 : I suggest to remove this sentence as there is no any loss without it. In general, make it more compact, and emphasize on the main findings and contributions."

We agree. It is removed. Conclusion is strongly shortened.

---

## Author Comment (AC2)

Referee #2

First of all we would like to thank the Referee for the large amount of work he has done revising out manuscript. In the revised version, we tried to follow his recommendations.

"The English text will be improved once the proposed corrections are accepted and performed."

The corrections suggested by Referee are introduced in the revised manuscript.

"Some citations are missing in various places and some others are proposed to be added (eg. lines 47, 81, 132, 224, 289)."

Citations are added

"It is not convincing that the detected aerosol layers correspond to smoke only, and not to other aerosol sources, as no air mass backtrajectory analysis with height has been shown. The authors shoud add these graphs in the supplement section."

The smoke layers were identified by the fluorescence capacity. For the episodes considered it exceeded $2.5*10^{-4}$, and no other aerosol type can provide it. The back trajectories for all considered cases show transport of smoke from North America (we added corresponding comment to the revised manuscript), but we would not like to overload the manuscript with additional figures.

"How convincing are the "*dry smoke events*" mentioned in the manuscript, when a complete air mass trajectory analysis is missing?"

In revised manuscript we removed "dry smoke layers" and use "layers with low water vapor content". Water vapor is measured by lidar. Besides, high values of the fluorescence capacity are possible only in the absence of hygroscopic growth.

**Specific comments**

"*line 51:* The paper of Wang et al., 2023, has to be omitted as it refers to a multi-wavelength elastic-Raman-fluorescence lidar system and not to a single-channel lidar as mentioned in the manuscript, as this placement there is missleading."

Done

"*line 119:* I would ask the authors to provide in a Supplement section a new figure detailing the transmission spectra (zoomin on the Transmission curves for T between 0-20%) of the filters at the 5 wavelengths mentioned, so that the readers can see (in detail) the overlapping transmission curves between these filters. Based on Veselovskii et al., (2023, Fig. 1) we can clearly see these overlapping regions, so the authors have to discuss on any induced errors in the detected fluorescence signals and their role on the accuracy of the retrieved aerosol parameters. "

Transmission bands of the interference filters are completely separated. Some overlap in Fig.1 (Veselovskii et al., 2023) corresponds to the transmission bands of the interference filter (IF) and the dichroic mirror (DM) used for separation: reflection of the DM starts to decrease near the long-wavelength edge of the IF. This effect is the strongest when 560 and 610 nm channels are separated, however it decreases the power at the 560 nm channel for less than 2%. This is

beyond accuracy of our calibration and so was ignored. In revised manuscript we added a phrase to the system secription.  Thus, probably no need to provide the Supplement section.

*"line 216*: See comments there (about the definition of the *nv* parameter etc.).*"*

Changed

*"page 21:* For clarity reasons limit the longitude up to 80° only (Fig. 2).*"*

Done

---

## Author Response (AR2)

Technical corrections in manuscript

We introduced all suggested corrections, except ln.339. We use $B$ because this is "spectral fluorescence backscattering coefficient", i.e. fluorescence backscattering per unit spectral range, while $\beta_F$ is fluorescence integrated over spectral interval.